# Analysis of the Conflict between Car Commuter’s Route Choice Habitual Behavior and Traffic Information Search Behavior

**DOI:** 10.3390/s22124382

**Published:** 2022-06-09

**Authors:** Kai Liu

**Affiliations:** School of Management, Nanjing University of Posts and Telecommunications, Nanjing 210023, China; liukai@njupt.edu.cn; Tel.: +86-15952035866

**Keywords:** route choice, habitual behavior, habit strength, traffic information, experiment

## Abstract

Motivated by the conflict between travelers’ habitual choice behavior and traffic information search behavior, in this paper, a behavioral experiment under different types of traffic information (i.e., per-trip traffic information and en-route traffic information) was designed to obtain data regarding car commuters’ daily route choices. Based on the observed data, participants’ route choices, habit strength, response time, and information search behaviors were analyzed. It is concluded that, in the beginning, the traffic information had a great influence on the habit participants’ route choices, let them think more, and made most of them switch from habit route to the best route (as recommended by traffic information); however, as time went on, the impact of traffic information declined, and several features of habits, such as automatically responding and repeated behavior, would reappear in some participants’ decision-making. Meanwhile, the different way of traffic information search behaviors (i.e., in active performance or in passive reception) could cause different information compliance ratios. These results would help to understand the interrelationship between car commuters’ daily route choice behaviors and traffic information search behaviors in short-term and in long-term, respectively, and provide an interesting starting point for the development of practical traffic information issuing strategies to enhance the impact of traffic information to alleviate traffic congestion during morning commuting.

## 1. Introduction

From the cognitive perspective, travelers’ route choice behaviors have been modeled in terms of either a perfect rational process or a boundedly rational process. The theory of expected utility maximization (EUM) is widely used for representing the perfect rational process [1]. Under this theory, researchers assume that the traveler chooses a route with the shortest (perceived) travel time or (perceived) travel cost [2,3]. On the other hand, the prospect theory [4], indifference band [5], and satisfying decision [6] are commonly used to represent the boundedly rational process, and these theories consider travelers’ cognitive limitations, intrinsic preferences, and information processing capability. While, under the assumption of perfect rational or boundedly rational process, many studies assume that travelers’ route choices are based on a relatively elaborate decision process, e.g., awareness, control, and efficiency [7,8,9,10]. However, does this relatively elaborate decision process always exist in travelers’ daily route choices?

In psychological research, many scholars find that, when a behavior is performed often in a relatively stable situation, a habit will be formed [11,12,13]. When the habit is formed, the decision-making process will be automated and performed without cognitive resource consumption [14,15,16,17]. Moreover, the more frequent a habit is being performed, the more automated and simpler the decision processes will be to use [11,18]. Especially, habit has been examined in travelers’ daily choices, for example, repeating a choice for one particular travel mode [19,20,21,22] or one particular travel route [23,24]. Habit can help travelers save cognitive resources, but it also presents a challenge for transport policy: even if travelers can be motivated to reduce their repeated choices, habitual choice tendencies are likely to nullify motivational gains [25]. Thus, for the transportation bureau, it is meaningful to consider how to effectively break travelers’ habitual choice behaviors.

In the studies of travel mode choice, researchers have conducted a lot of studies to investigate the effects of free tickets [26], residential relocation [27], life events [28], spatial context [22], environmental attitude [16], additional incentives [29], and traffic information [11] on travelers’ daily mode choice habitual behaviors. Nevertheless, one is forced to recognize that traffic information is the most pervasive. Then, can traffic information significantly influence travelers’ daily mode choice habitual behaviors? Verplanken et al. [11] showed that the more habitual the driving choice is, the less information is searched. Aarts et al. [30] found that strong-habit drivers searched for less information than weak-habit drivers. Moreover, similar results can also be found in the literature [22]. In addition, Anderson and Lebiere [31] suggested that, during the repeated decision-making process, travelers may gradually reduce their focus on information about choice situations and choice options and then establish and use simple rules to make decisions. At this point, we can see that there is a conflict between habit travelers’ daily mode choices and their traffic information search behaviors.

While, in the studies of travel route choice, much attention has been paid to the traffic information, they have also found that the types of traffic information [10] and the accuracy of the traffic information [32] have a great influence on travelers’ route choice behaviors. In addition, Srinivasan and Mahmassani [33] found that, under the impact of real-time information, travelers’ route choice mechanisms can be divided into inertia and compliance. Avineri and Prashker [34] suggested that informed travelers choose more reliable routes to avoid risk. However, as mentioned above, since habit exists in travelers’ daily route choices, then does the conflict also exists in habit travelers’ daily route choices and their traffic information search behaviors?

To answer the above question, this paper focuses on the car commuters’ daily route choices (for most car commuters, the daily commute happens in a relatively stable situation with a fixed origin-destination pair, fixed stare working time, and similar road condition, if the extreme weather, traffic accident, and life events are not considered), and explores the conflict between car commuters’ daily route choice habitual behaviors and traffic information search behaviors. To our best knowledge, in terms of focus on the psychology of the habit (e.g., automatically responding, repeated behavior, and without cognitive elaboration), little research has shed light on the relationship between habit travelers’ route choices and their traffic information search behaviors or exposed the impact of traffic information on habit travelers’ daily route choices in the short-term and in long-term, respectively.

Moreover, in this study, a behavioral experiment under different types of traffic information (e.g., pre-trip traffic information and en-route traffic information) was designed to obtain data regarding commuters’ daily commuting decision-making, and based on the empirical study, we aimed to: (i) examine car commuters’ daily route choice habitual behaviors and measure their route choice habit strength; (ii) investigate the habit commuters’ traffic information search behaviors in the short-term and long-term, respectively; (iii) study the psychology of habit commuters with and without traffic information; (iv) analyze the influence of the different ways of traffic information search behaviors (i.e., in active performance vs. in passive reception) on habit commuters daily route choices. The research results in this paper will help to understand the interrelationship between habit commuters’ daily route choice and their traffic information search behaviors in the short-term and in long-term, respectively, and will provide an interesting starting point for the development of practical traffic information issuing strategies to enhance the impact of traffic information to alleviate traffic congestion during morning commuting.

The rest of the paper is arranged in five sections. Section 2 defines the theoretical model and proposes several hypotheses. Section 3 describes the design of the experiment as well as the data collection procedures. Section 4 presents the longitudinal analysis of the collection data and further compares the difference between the impact of pre-trip traffic information and en-route traffic information on habit commuters’ route choices. The results of the experiment are discussed in Section 5. Section 6 offers a conclusion.

## 2. Theories and Methods

Habit has been commonly found in travelers’ daily route choice behaviors. Figure 1 gives clear evidence (based on the statistical results of the plate recognition date), since in the behaviorist tradition, habit strength has been considered to be a function of repetition [12,31]. Then, as shown in the figure, different travelers have different habit strength. However, as suggested by some psychological research [11,22], the higher the habit strength is, the less information the traveler will need. Thus, what is the relationship between the car commuters’ route choice habit strength and traffic information search behaviors?

To answer this question, the first thing to realize is that there are many types of traffic information in reality, and different types of traffic information may have different effects [9,35,36]. For example, Parvaneh et al. [37] suggested that the pre-trip and en-route traffic information has different impacts on drivers’ decision-making. Ji et al. [38] suggested that traffic radio prevents drivers from encountering the serious delays, while VMS has the highest discrete degree but accuracy in travel time prediction. Long et al. [10] compared four different types of traffic information on commuting travelers’ route choices, i.e., historical descriptive information, real-time descriptive information, historical travel time information, and real-time travel time information, and found that the real-time travel time information had the greatest impact on an individual switching behaviors.

The second thing to realize is that the different ways of travelers’ information search behaviors, e.g., in active performance or in passive reception, may also have a great impact on their decision-making. This is because seeking traffic information in active performance will cost more cognitive resources, and this conflicts with the psychology of habit (i.e., without cognitive elaboration). While seeking traffic information in passive reception will cost less cognitive resources, this is relatively consistent with the psychology of habit. Therefore, to further examine the relationship between car commuters’ route choice habit strength and traffic information search behaviors, an experimental study was used for this paper. Two types of traffic information (i.e., pre-trip traffic information and en-route traffic information) were considered, and we assumed that car commuters seek pre-trip traffic information in active performance and seek en-route traffic information in passive reception. Now, before the empirical study, the following hypothesizes are given.

**H1.** *Higher level of car commuters’ route choice habit strength correlates with a lower level of traffic information needed, and vice versa*.

**H2.** *Seeking traffic information in active performance has a lower influence on habit commuters’ route choices, and vice versa*.

**H3.** *Seeking traffic information in passive reception has a higher influence on habit commuters’ route choices, and vice versa*.

Now, according to the above hypothesizes, it is necessary to consider how to measure a car commuter’s habit strength. As mentioned above, in the behaviorist tradition, researchers consider habit strength to be a function of repetition [12,30]. This means that the more frequent one particular option is being repeatedly chosen, the higher the habit strength is. However, as suggested by Jager [39], “it is not the frequency of behavior that determines the strength of a habit, but the degree to which the behavior has been automated and is being performed without cognitive elaboration”. Then, for the sake of accuracy, in this study, the self-report habit index (SRHI) [18] was used to measure car commuters’ route choice habit strength.

The SRHI relies on individual’s experiences of repetition and automaticity and can be used to assess an individual’s subjective experience of several features of habits. The individual responds to a stem (‘Behavior X is something…’), which contains 12 SRHI items, and it requires them to reflect on the lack of awareness, automaticity, lack of control, and mental efficiency of a given behavior. The responses were made on a 7-point Likert-type scale ranging from 1 (agree) to 7 (disagree). The scores were recoded such that high values indicated strong habit. The SRHI has been shown to have good test-retest reliability and also has been used to predict behavioral frequency in many domains, including eating behaviors, physical activities, and travel mode choices (for reviews, see the literature [40]).

Note that the SRHI uses individual’s self-reported data to measure habit strength. However, when recalling the past travel experiences, the individual may be affected by some biases such as availability or representativeness [41]. This will make some individuals’ self-reported data deviate from their actual choices. For the sake of accuracy, the response frequency measure (RFM) [11] was used to test car commuters’ route choices habit strength, which are measured by using SRHI. The RFM relies on the frequency of choosing one particular option over a period of time serving as a measure of habit strength. In general, a higher value of RFM means a higher score of SRHI.

## 3. Case Study

### 3.1. Experimental Design

The experiment consisted of four procedures. Figure 2 gives the relationship among each procedure. Firstly, the experiment was conducted with 30 trials to simulate car commuters’ daily route choices without any traffic information except feedback information of the chosen route’s actual arrive time (note that the first 10 choices were used as warm ups). Secondly, an SRHI survey questionnaire was presented to the participants to investigate their route choice habit strength in the first procedure (mainly focused on trials 11–30). Thirdly, the experiment was continued with 20 remaining trials, but participants were provided with two types of traffic information: pre-trip traffic information (provided in trials 31–50) and en-route traffic information (provided in trials 41–50). It is worth mentioning that participants seek pre-trip traffic information in active performance (e.g., predicted traffic information, which can be obtained by using a smart phone to search for relevant information), while they seek en-route traffic information in passive reception (e.g., variable message sign (VMS), which can be noticed by drivers if they pass by it). Finally, another SRHI survey questionnaire was presented to participants to investigate their route choice habit strength in the remaining 20 trials.

We used Z-tree as the experimental platform [42]. During the experiment, participants were required to submit their decision-making process and choices in each trial. The road network, which was used in the experiment, is shown in Figure 3. There was one Origin-Destination (OD) pair with five links and three routes. M and N are nodes in the road network. Route A and Route B were described as urban roads and had a common link, i.e., link 1. Route C was described as a highway-ring around the city. Note that, in this study, we mainly focused on the individual’s route choice behavior, so the interaction between participants was not considered. Moreover, the links’ actual travel times were randomly drawn from five independent normal distributions, as shown in Table 1.

Participants were informed that traffic congestion only occurs on link 2, link 3, and link 5, and it made these links’ actual travel times uncertain. Moreover, participants were also told that the traffic congestion on link 3 was heaviest, then link 2, and finally link 5. Descriptive information—predicted travel time on each route—was used as the traffic information and was provided to participants only in the remaining 20 trials. The predicted travel time was obtained based on the actual travel time, which was drawn from Table 1, but had some deviation because of the uncertainty. In the experimental design, there was a 10% random deviation between the predicted travel time and the actual travel time. Note that en-route traffic information was only provided at the node N, such as VMS, which tells participants who pass by node N the predicted travel time on link 2 and link 3.

### 3.2. Participants and Experimental Procedures

An ad was posted at a residential area near the university from 1 July through 31 August 2019. People who commuted by private cars at least twice a week were recruited. Since, in our experiment, participants’ interactions were not considered, people could participate in the experiment at any time they preferred within the agreed upon two-month experimental window. A total of 57 participants finished the whole experiment during the two-month experimental window. Before the experiment properly started, participants were asked to finish a questionnaire that included demographic characteristics, risk attitudes, daily route choice, and departure time setting, etc. Table 2 presents the main characteristics of the participants.

At the beginning of experiment, participants were preliminarily instructed about the experiment. The length and the type of all available routes were described to participants before starting the task. The task was described as making a series of 50 route choices, and in the first 30 trials, there was no traffic information provided. In the remaining 20 trials, participants could receive traffic information. Note that the pre-trip traffic information was provided in trials 31–50, and the en-route traffic information was provided in trials 41–50. In order to encourage realistic behavior, participants were asked to imagine that their start working time was 9:00 a.m., and they could choose to depart freely at any time they wished, but lateness was not allowed. After each round of choice, feedback information of the chosen route’s actual arrive time was provided to each participant.

In addition, when participants finished the first 30 trials, they could receive a five-minute break. Subsequently, an SRHI survey questionnaire concerning route choices in trials 11–30 was presented to each participant, and the responses were made on a 7-point Likert scale ranging from 1 (agree) to 7 (disagree). When participants finished the SRHI, there was another five-minute break. After this break, participants got to the remaining 20 trials. In these trials, traffic information was provided. Note that the pre-trip traffic information was not directly provided to participants, and if a participant wanted pre-trip traffic information, she/he must click the button on the screen to obtain the information. After participants finished the remaining 20 trials, there was also a five-minute break; then, another SRHI survey questionnaire concerning route choices in the remaining 20 trials was presented to them.

Finally, at the end of the whole experiment, each participant would receive a show-up fee of CNY 80 and a best CNY 80 reward if she/he were never late for work in each trial and their route choice habit strength, whether it came from SRHI or RFM, remained at the same level (i.e., a higher value of RFM means a higher score of SRHI).

### 3.3. Data Collection and Measurements

The whole experiment included making a series of 50 choices by filling in the boxes and selecting from a list on the screen (see Figure 4). The data of participants’ departure times, route choices, expected arrive times, actual arrive times, and response times (i.e., the time spending on each trial of decision-making and choices), etc., were recorded in the Z-tree. Moreover, as mentioned in the experimental design, since individuals seek the per-trip traffic information in active performance, in trials 31–50, there was a “traffic information” button on the bottom left-hand corner of the screen, and only when participants clicked this button, the pre-trip traffic information would be shown. Therefore, the number of times a participant clicked the button were also recorded in the Z-tree. In addition, in trials 41–50, after participants finished the route choices and clicked the “Ok” button on the bottom right-hand corner of the screen, the en-route traffic information about predicted travel time on link 2 and link 3 were shown on the screen to those participants who chose Route A or Route B as their commuting routes. Then, those participants needed to choose link 2 or link 3 to finish their trip. Note that the en-route traffic information was displayed for only 10 s, and if the timeout happened then the en-route traffic information would disappear.

The design of SRHI is shown in Table 3. It is worth mention that, in our experiment, participants should have finished two SRHI survey questionnaires concerning route choices with and without traffic information, respectively. In these two SRHIs measures, participants needed to respond to the stem ‘Choosing route X in trials 11–30 (i.e., without traffic information) or in remaining 20 trials (i.e., with traffic information) is something…’, and were presented with the 12 SRHI items. Responses were made on a 7–point Likert-type scale ranging from 1 (agree) to 7 (disagree). Note that the scores were recoded such that the high values indicated strong habits [18,40].

## 4. Results

### 4.1. Longitudinal Analysis

The self-report habit index (SRHI) and the response frequency measure (RFM) were used to describe participants’ route choice habit strength in the experiment, respectively. Verplanken et al. [11] defined response frequency as the frequency of choices for one particular travel mode within a certain period of time. In this study, we used this definition, and the participant’s response frequency measure (RFM) could be written as:RFM = The number of times a route was chosen/the number of trials.(1)

Considering that, at the start of the experiment, due to the participants not being familiar with the simulated commuting environment, the elaborate decision processes would exist in their choices. This was against the feature of habit. Thus, for the sake of accuracy, the first 10 trials were perceived as warm ups, and data analysis mainly focused on trials 11–50. Figure 5 shows the statistical results of participants’ route choice habit strength in trials 11–30 and in trials 31–50 by using the measure of SRHI and RFM, respectively. It is worth noting that, if a participant’s route choice habit strength, whether it came from SRHI or RFM, did not remain at the same level, then the collection data of this participant was discarded. Finally, two thousand one hundred and twenty observations of 53 participants were obtained.

We considered the frequency of choices for one particular route more than half in trials 11–30 or in trials 31–50. As shown in Figure 5, in trials 11–30 (without traffic information, see left figure), the habit existed in some participants’ route choices, and there were 22 participants who chose one particular route more than sixteen times in trials 11–30. In addition, participants had different habit strength, and the larger the value of the RFM, the higher the score of the SRHI. For the sake of convenience in researching the problems, based on values of SRHI and RFM, we defined SRHI ≥ 6 as the strong habit, 4 ≤ SRHI < 6 as the intermediate habit, 2 ≤ SRHI < 4 as the weak habit, and SRHI < 2 as no habit. Thus, the habit participants included strong habit participants, intermediate habit participants, and weak habit participants.

On the flip side, in trials 31–50, when traffic information was provided, habit became weak, and only two participants chose one particular route more than sixteen times. This result indicates that traffic information can effectively influence many habit participants’ route choices. Specifically, this can be further verified by habit participants’ response times in each round of trials (see Figure 6). As shown in Figure 6, after the warm-ups (i.e., trials 1–10), habit participants’ average response times significantly decreased in trials 11–30. This is because, with the increasing familiarity with the simulated commuting environment, repeating a choice in a relatively stable situation made some participants gradually accumulate travel experiences and develop habits. Then, lack of awareness and automatically responding made these habit participants think less. Subsequently, when traffic information was provided, the habit participants’ average response times significantly increased in the first few trials. This result indicates that traffic information can make habit participants think more.

As the experiment went on, the average response time gradually decreased (showed a small increase when en-route traffic information was beginning to be provided to participants), and in the last few trials, the average response time had minor changes (see Figure 6, trials 31–50). This suggests that habit participants were sensitive to new traffic information and adopted a mind-set that was conducive to behavior change [13]. However, along with the deepening of traffic information, the information understanding of participants gradually changed from a strange familiarity and understanding. This weakened the role of traffic information and made some participants reduce dependence on it. The above analysis results can be further verified by the number of times the pre-trip traffic information was used by habit participants in trials 31–50 (see Figure 7).

In addition, when traffic information was provided, there were still two strong habit participants. Then, by checking these two participants’ pre-trip traffic information usages in trials 31–50, we found that strong habit participants searched less information than other types of habit participants. One reason may be that the differences of predicted traffic times among the three routes were not huge (see Figure 8), and compared to the information recommended route (which was not equal to the habit choice route), a little bit more travel time on the habit choices route can be accepted by these two strong habit participants. Another reason may be that, within a few tries, strong habit participants were more certain about the routes’ predicted travel times that they did not inspect, then to save cognitive cost, they searched less information in the following tasks (see Figure 9).

Finally, to identify significant differences in participants’ route choice habitual behaviors, an analysis of variance (ANOVA) was conducted. A separate analysis was also conducted for no information trial (trials 11–30), information trial (trials 31–50), information trial period one (trials 31–40, denoted as information-1), and information trial period two (trials 41–50, denoted as information-2). Results of the ANOVA for within (F-test) and between-group (*t*-test) significance are depicted in Table 4. As expected, for strong habit participants, the score of the SRHI, response time, and pre-trip traffic information usage were significantly different from other participants, i.e., the strong habit participants used the simple rule to make a choice and searched less traffic information. For weak habit participants and no habit participants, the behavior with or without traffic information was similar in response time and information usage.

Behavior with traffic information, in trials 31–40 vs. in trials 41–50, was also quite different. First, habit participants’ response times were significantly higher in trials 31–40. Second, the use of pre-trip traffic information was significantly lower in trials 41–50, except for intermediate habit participants. These results suggest that, in the long run, as participants gain more experiences, the impact of pre-trip traffic information decreased. Then, even with traffic information, as time goes on, several features of habits, such as automatically responding and repeated behavior, will be back. This has also been found in travelers’ travel mode choices [22]. The above ANOVA results further verify that there is a conflict between car commuters’ route choice habitual behaviors and traffic information search behaviors, and a higher level of car commuters’ route choice habit strength correlates with a lower level of traffic information needed. Thus, the assumption of H1 is tenable.

### 4.2. Pre-Trip Traffic Information vs. En-Route Traffic Information

In order to further analyze the difference between the pre-trip traffic information and en-route traffic information, the present study employed the concept of compliance. Srinivasan and Mahmassani [33] defined compliance as the tendency of a traveler to comply with the recommended best route (as suggested by traffic information). Then based on the observation data, we used this definition to compute the habit participants’ compliance ratios under the pre-trip traffic information and en-route traffic information, respectively. For the sake of accuracy, the compliance ratio under pre-trip traffic information was restricted to these participants who used the traffic information.

The habit participants’ compliance ratio (HC) was computed as:(2)$$HC=\sum_{K}mc_{k}/\sum_{k}m_{k},\;k=31,\;32,\ldots ,50$$
where *k* is a generic round, $mc_{k}$ is the number of compliant habit participants at round *k*, $m_{k}$ is the number of habit participants who used traffic information at round *k*. Then, the compliance ratios under pre-trip traffic information and en-route traffic information were calculated and are shown in Figure 10. It is worth mentioning that, when pre-trip traffic information and en-route traffic information were provided at the same time, due to some trials, the best route that was recommended by pre-trip traffic information and en-route traffic information was not same; thus, for the sake of accuracy, the compliance ratios under the pre-trip traffic information in trials 41–50 were further restricted to the participants who sought traffic information in active performance and chose the recommended best route at the very beginning of the trip.

As seen in Figure 10, no matter if they are under the pre-trip traffic information or under the en-route traffic information, the compliance ratios were not always equal to 100%. One reason may be that, in some trials (e.g., trials of 35 and 36), the recommended best route (as suggested by pre-trip traffic information) was Route C; however, in some habit participants’ daily route choice sets, there were only Route A and Route B. This means that those habit participants only considered Route A or Route B as their daily commuting routes. Specially, Zhang and Yang [43] defined this choice behavior as route choice inertia with a prevailing choice set. Another reason may be that, in some trials, the differences of the predicted route travel times among the three routes (predicted by per-trip traffic information) or the difference of the predicted link travel times between two links (predicted by en-route traffic information) were not significant. As proposed by the literature [44,45,46], there is an “indifference band” that exists in the traveler’s route choice behavior, and if the difference between travel times on the current route and on the best available route is within this indifference band, the traveler will not switch a route. Therefore, the compliance ratios were not always equal to 100% under the influence of traffic information.

Then, from the holistic perspective, the habit participants’ compliance ratios under the en-route traffic information were smaller than the compliance ratios under the pre-trip traffic information. This suggests that pre-trip traffic information has a stronger effect on the habit participants’ route choices than the en-route traffic information. Considering this, participants sought pre-trip traffic information in active performance and sought en-route traffic information in passive reception. In addition, due to pre-trip traffic information, this can make habit participants think more (see Figure 6 and ANOVA results in Table 4). Thus, it was concluded that traffic information that has to be actively acquired rather than passively acquired is more influential on habit commuters’ decision-makings. To further confirm whether there is a significant difference between the compliance ratio under the pre-trip traffic information and the compliance ratio under the en-route traffic information, an ANOVA was conducted [F(1,28) = 4.232, *p* = 0.049 < 0.05]. The post-hoc comparison showed a statistically significant difference of the compliance ratio between pre-trip traffic information and en-route traffic information. Finally, the above analysis results suggest that seeking traffic information in active performance will have a greater impact on commuters’ route choice habitual behaviors than seeking traffic information in passive reception. Therefore, the assumptions of H2 and H3 are not tenable.

## 5. Discussion

A review of this study showed that, in a stable context, when a decision behavior is being performed daily, habit will be formed [12,13,30], and the more frequent a habit is being performed, the less the information is needed [13,30,47]. Thus, motivated by the conflict between individuals’ habitual choices and their information search behaviors, this study suggested a need to examine the impact of traffic information on car commuters’ daily route choice habitual behaviors. A theoretical design covering both methods and hypotheses has been developed for examining. Notably, with the empirical study, this study successfully showed the conflict between car commuters’ route choice habitual behaviors and information search behaviors, e.g., traffic information can reduce the car commuters’ route choice habit strength and make them break habit, but as time goes on, several features of habits, such as automatically responding and repeated behavior, will reappear in some car commuters’ daily route choices. Now, based on the research results in this paper, we created the following discussions.

Firstly, in this study, we used a self-report habit index (SRHI) and response frequency measure (RFM) to measure participants’ route choice habit strength, and found out that different participants have different habit strengths. In addition, traffic information can reduce participants’ route choice habit strength and make them willing to break habit. However, one result of this study also showed that, when traffic information was provided, there were still existing strong habit participants, even if their habitual choice routes were not the best (as recommended by traffic information). These findings are meaningful because they indicate that traffic information has significant influence on the car commuters’ route choice habitual behaviors, but there also exists a part of car commuters whose daily route choices are not dependent on the traffic information. Although we used the concepts of “prevailing choice set” and “indifference band” to explain why some habit participants reject the traffic information. Jager [39] indicated that the conflict between a habit and new information would cause cognitive dissonance, which can be resolved by trivializing or rejecting the information, and trivialization or rejecting dissonant information may be a lot easier than actually changing one’s habit. Thus, further research can be conducted from different angles to explain why some habit drivers reject the traffic information.

Secondly, under the influence of pre-trip traffic information, as suggested by our research results, in the first few trials, it played an important role and let habit participants think more and made some of them break their habitual choices. However, as time went on, the impact of pre-trip traffic information was gradually decline. Automatically responding and repeated behavior reappeared in some habit participants’ route choices. This finding is important because it indicates that, in the long run, as car commuters become well acquainted with the pre-trip traffic information, they are more certain about the attribute values of all available routes, even if they do not inspect traffic information. This will gradually reduce the car commuters demand for the pre-trip traffic information and finally make them reuse simple rules to make decisions. Thus, if transportation researchers want to evaluate the impact of pre-trip traffic information on car commuters’ daily route choices in the long run, they must consider the decline effect of traffic information. Moreover, further efforts can be made toward characterizing the decline effect of traffic information by using a function.

Thirdly, in our experimental design, the habit participants sought pre-trip traffic information in active performance and sought en-route traffic information in passive reception. Compared to the en-route traffic information, the pre-trip traffic information made habit participants think more. In addition, under the influence of pre-trip traffic information, the habit participants’ average compliance ratio was higher. These findings are also meaningful because they indicate that traffic information that has to be actively acquired rather than passively acquired is more influential on habit traveler’s decision-making. Moreover, the ANOVA results also suggest that, for the strong habit participants, the score of the SRHI, response time, and pre-trip traffic information usage were different from other types of habit participants. These findings further verify that there is a conflict between car commuters’ route choice habitual behaviors and traffic information search behaviors; and the higher the travelers’ habit strength is, the less the traffic information is needed.

Fourthly, in our experiment, all participants who chose link 1 as their first travel link received and noticed the en-route traffic information at the end of the link. However, it is unreasonable to assume that all drivers will always notice the en-route traffic information, i.e., VMS, because sometimes vehicle speed is faster and it requires drivers to switch their attentions from the front windshield to the VMS in active performance. Then, it is possible that the en-route traffic information, similar to VMS, may be less likely to be noticed by drivers, that is, the compliance ratio under the en-route traffic information may be much smaller. Thus, further research can explore drivers’ notices on en-route traffic information, i.e., VMS, by designing an eye movement experiment.

Fifthly, in our study, we calculated the compliance ratio based on the Srinivasan and Mahmassani [33] definition, that is, if a traveler’s choice route is the best route (as recommended by traffic information), she/he is in compliance with traffic information. However, as suggested by Ben-Elia et al. [32], “the compliance is a latent construct and cannot observed directly whether a traveler complies with traffic information suggestions or chooses the information recommended route because that is what she/he thinks is the best alternative, independently of traffic information”. Therefore, in our study, the compliance ratios under the pre-trip traffic information and en-route traffic information were higher than they actually would be. Thus, further research can be made toward how to efficiently calculate the habit travelers’ traffic information compliance ratio.

Finally, as time goes on, the effect of traffic information declines; then, it is necessary to consider the impact of transport infrastructure (i.e., VMS) and traffic regulation measures on flexibility traffic users’ route choices in the short-term and in long-term, respectively. Moreover, it would also be interesting to analyze the influential parameters (driver’s sense of safety, complexity of intersections, traffic regulation, road slopes in winter conditions, etc.) and develop a model to predict drivers’ route choices with the traffic information in real traffic conditions.

## 6. Conclusions

This study focused on the conflict between car commuters’ route choice habitual behaviors and traffic information search behaviors. An experiment was conducted to obtain participants’ route choice data as well as traffic information search data. Then, different methods were applied to deal with the experimental data and came to the following conclusions: (1) at the beginning, the traffic information had great influence on car commuters’ route choice habitual behaviors; but as time went on, the effect of traffic information was on the decline; (2) the different ways of traffic information search behaviors (i.e., actively or passively), could have different influences on habit participants’ response times and could cause different information compliance ratios; (3) habit will reappear in car commuters’ daily route choices when they become well acquainted with the traffic information; (4) policies for evaluating the impact of traffic information should consider commuters’ habitual behavior and their traffic information search behaviors and should focus on both the short-term and long-term.

It is worth mentioning that due to the difference between the experimental design and an actual situation, the scope of this study was restricted to an awareness of the conflict between car commuters’ route choice habitual behaviors and traffic information search behaviors. Even so, there are also some limitations in this study; for example, the measure of the traffic information’s compliance ratio is preliminary, and this makes the compliance ratio under the en-route traffic information higher than it may be in reality; moreover, the compliance ratios under the pre-trip traffic information and en-route traffic information were also higher than they actually would be. Thus, in future research, it is necessary to build a reasonable and effective method to calculate drivers’ information compliance ratios.

## Figures and Tables

**Figure 1 sensors-22-04382-f001:**
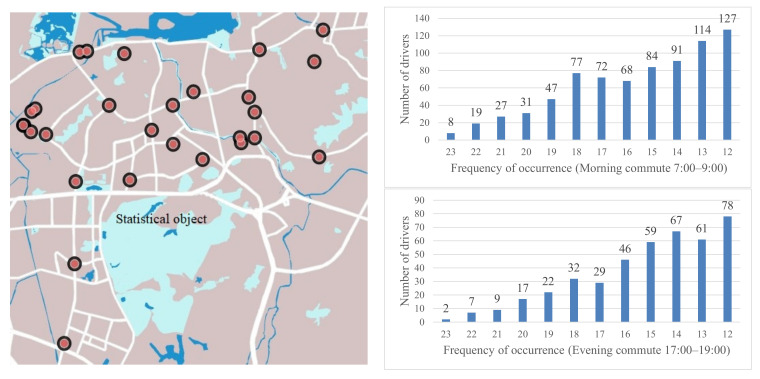
Evidence of habit from the statistical results of the plate recognition date. (The left figure shows the distribution of 27 road securities (see the red circles). The right figure shows the statistical results of the frequency of the same car that showed up at the same road security from 20 March 2018 to 16 April 2018, for 23 workdays. There are about 33,488 observation dates in the morning and 33,258 observation dates in the evening.)

**Figure 2 sensors-22-04382-f002:**
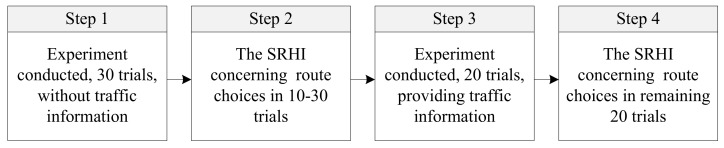
The design of the experimental procedure.

**Figure 3 sensors-22-04382-f003:**
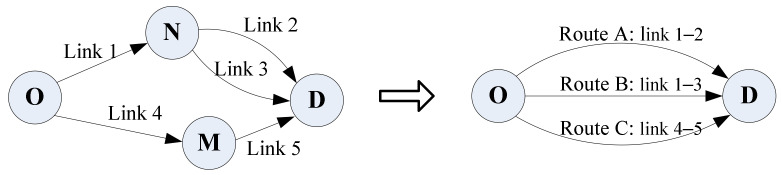
Network used in the experiment.

**Figure 4 sensors-22-04382-f004:**
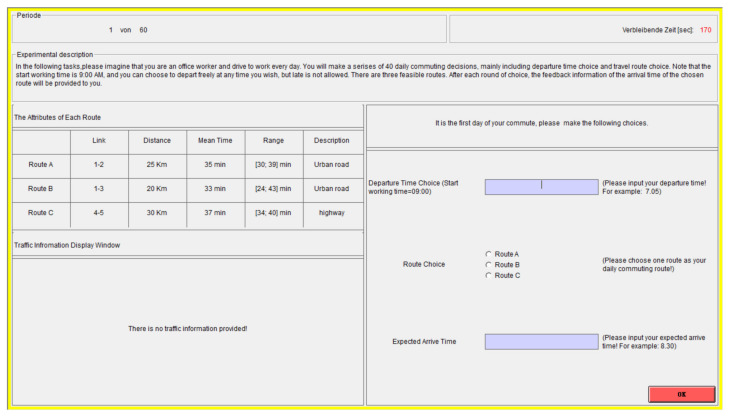
Snapshot of the simulation window (translated from Chinese).

**Figure 5 sensors-22-04382-f005:**
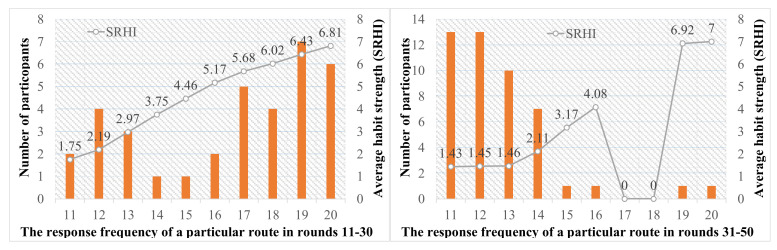
The statistical results of participants’ route choice habit strength.

**Figure 6 sensors-22-04382-f006:**
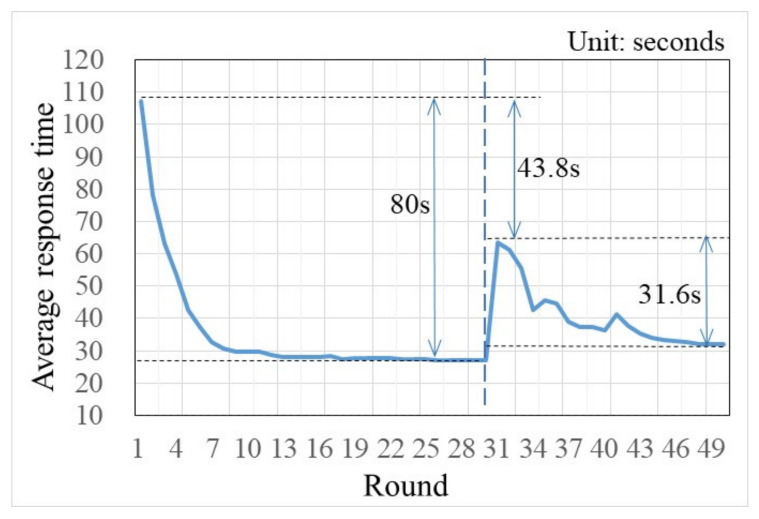
The average response times in rounds 1–50.

**Figure 7 sensors-22-04382-f007:**
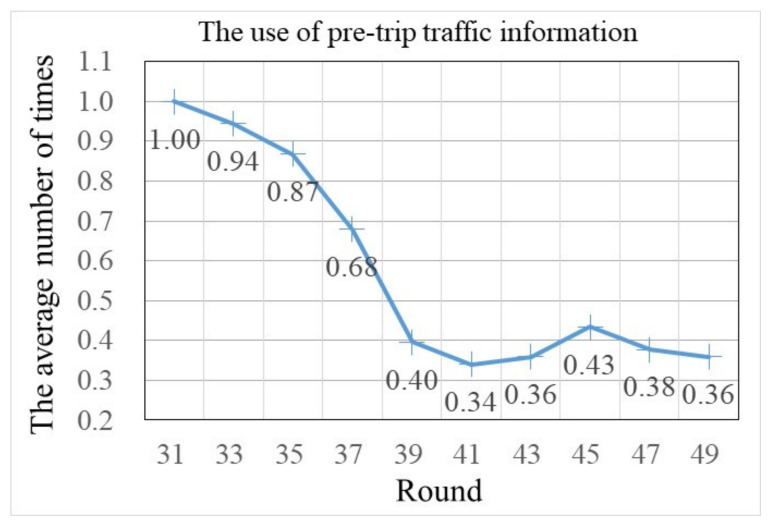
The average number of times the pre-trip traffic information was used in rounds 31–50.

**Figure 8 sensors-22-04382-f008:**
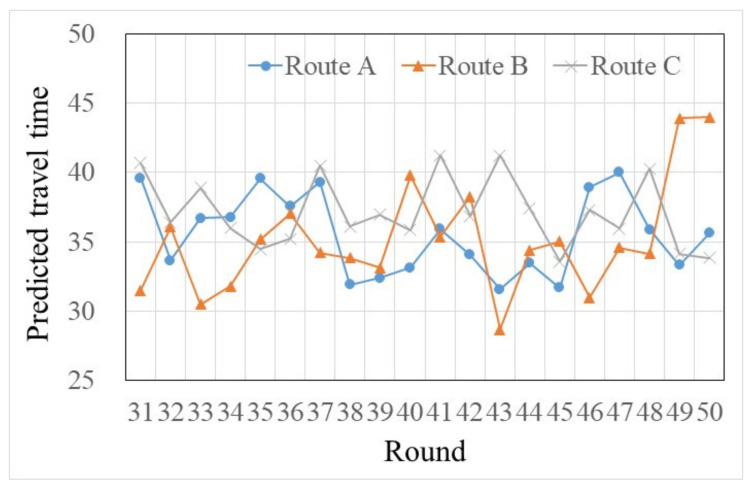
The travel time predicted by pre-trip traffic information in rounds 31–50.

**Figure 9 sensors-22-04382-f009:**
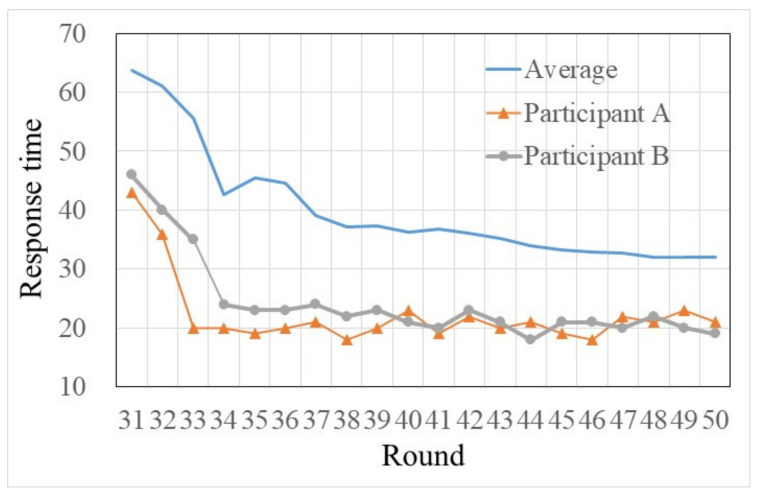
The response time of two strong participants in rounds 31–50.

**Figure 10 sensors-22-04382-f010:**
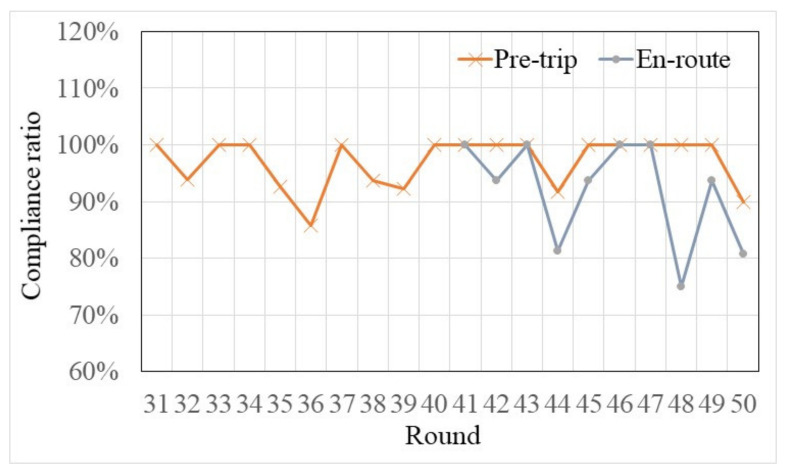
The compliance ratio under the pre-trip and en-route traffic information. (The accuracies of pre-trip traffic information and en-route traffic information were both 90%, i.e., there was a 90% chance that the traffic information recommended the best route equal to the actual result.)

**Table 1 sensors-22-04382-t001:** Distribution of link travel times.

Link	Distance (km)	Mean (min)	Variance	Range (min)
1	12	15	0.1094	[14; 16]
2	13	20	1.3715	[16; 23]
3	8	18	7.9859	[10; 27]
4	25	30	0.1094	[29; 31]
5	5	7	0.4390	[5; 9]

**Table 2 sensors-22-04382-t002:** Description of participants.

Variable	Type	Frequency
Sex	Male	63.16%
	Female	36.84%
Age	25–30	47.37%
	31–40	31.58%
	41–60	21.05%
Education attainment level	Bachelor’s degree	59.65%
	Master’s degree	26.32%
	PhD	8.77%
	Secondary school	5.26%
Occupation	University lecturers/researchers/teachers	15.79%
	Office workers/salesperson/service workers	54.39%
	Civil service/hospital workers/others	29.82%

**Table 3 sensors-22-04382-t003:** The self-report habit index (SRHI).

Choosing Route X in Trials 11–30 (or in Trials 31–50) Is Something…
1. I do frequently.
2. I do automatically.
3. I do without having to consciously remember.
4. That makes me feel weird if I do not do it.
5. I do without thinking.
6. That would require effort not to do it.
7. That belongs to my daily routine.
8. I start doing before I realize I’m doing it.
9. I would find hard not to do.
10. I have no need to think about doing.
11. That’s typically ‘me’.
12. I have been doing for a long time.

**Table 4 sensors-22-04382-t004:** ANOVA results (between and within group differences).

Scenario	Maximization Score of SRHI (Average Score of SRHI)									
	Strong (S)	Intermediate (I)	Weak (W)	No (N)	t-Value (Sig.)					
					S and I	S and W	S and N	I and W	I and N	W and N
No information	7.00 (6.53)	5.92 (5.24)	3.75 (2.81)	1.83 (1.49)	8.13 **	21.04 **	58.93 **	13.32 **	17.89 **	6.22 **
Information	7.00 (6.96)	4.08 (4.08)	2.58 (2.44)	1.92 (1.43)	_ ^b^	16.02 **	67.75 **	_ ^b^	_ ^b^	4.97 **
F-value	3.42 *	3.59 *	0.80 ^a^	0.47 ^a^						
**Scenario**	**Maximization Response Time (Average Response Time)**									
	**Strong (S)**	**Intermediate (I)**	**Weak (W)**	**No (N)**	**t-Value (Sig.)**					
					**S and I**	**S and W**	**S and N**	**I and W**	**I and N**	**W and N**
No information	44 (24)	47 (26)	61 (27)	60 (28)	−1.90 *	−3.95 **	−4.19 **	_ ^a^	−3.83 **	_ ^a^
Information	46 (23)	61 (39)	82 (42)	91 (40)	−8.50 **	−14.32 **	−16.06 **	−2.50 **	_ ^a^	_ ^a^
F-value	_ ^a^	53.800 **	36.825 **	349.14 **						
Information-1	46 (26)	61 (45)	82 (48)	91 (46)	−6.94 **	−9.45 **	−17.10 **	_ ^a^	_ ^a^	_ ^a^
Information-2	23 (21)	40 (33)	41 (35)	45 (34)	−6.82 **	−23.00 **	−10.57 **	−1.86 *	_ ^a^	_ ^a^
F-value	7.80 **	8.64 **	51.78 **	511.59 **						
**Scenario**	**Total Number of Pre-Trip Traffic Information Usage (Average Rate)**									
	**Strong (S)**	**Intermediate (I)**	**Weak (W)**	**No (N)**	**t-Value (Sig.)**					
					**S and I**	**S and W**	**S and N**	**I and W**	**I and N**	**W and N**
Information	6 (0.15)	11 (0.55)	58 (0.58)	577 (0.55)	−3.94 **	−5.15 **	−6.62 **	_ ^a^	−3.94 **	_ ^a^
Information-1	5 (0.25)	7 (0.70)	39 (0.78)	379 (0.73)	−3.00 **	−3.56 **	−5.94 **	_ ^a^	−1.96 *	_ ^a^
Information-2	1 (0.05)	4 (0.40)	19 (0.38)	198 (0.38)	−2.45 **	−3.68 **	−3.68 **	_ ^a^	−3.67 **	1.94 *
F-value	3.23 *	_ ^a^	19.25 **	145.08 **						

** Significant at 0.05; * significant at 0.1; ^a^ not significant; ^b^ the correlation and t cannot be computed because the sum of case weights is less than or equal to 1.

## Data Availability

Not applicable.

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
