# Peer review of "Analysis of the Conflict between Car Commuter’s Route Choice Habitual Behavior and Traffic Information Search Behavior"

_sensors, 2022, doi:10.3390/s22124382_

Round 1

Reviewer 1 Report

The authors presented a very interesting paper related to commuters’ route choice habitual behavior and traffic information search behavior.

1. The authors can consider adding more recent related publications related to commuters’ route choice behavior: “Impacts of Pokémon GO on the route and mode choice decisions: exploring the potential for integrating augmented reality, gamification, and social components in mobile apps to influence travel decisions”, “Predicting pedestrian flow along city streets: A comparison of route choice estimation approaches in downtown San Francisco”, “Paving the Way for Autonomous Vehicles: Understanding Autonomous Vehicle Adoption and Vehicle Fuel Choice under User Heterogeneity” and other 2021 and 2022 publications.

2. The authors mentioned in detail in terms of the experiment procedure and participants’ characteristics. However, it is not clear how many participants are there.

3. What are the limitations of this study?

Author Response

Thanks for the suggestion. In the revision paper, I have added the following contents in introduction section, and updated the References:

In the studies of travel mode choice, researchers have conducted a lot of studies to investigate the effects of free tickets [26], residential relocation [27], life events [28], spatial context [22], environmental attitude [16], additional incentives [29], and traffic information [11] on travelers’ daily mode choice habitual behaviors.

  1. Hasnine, M.S.; Habib, K.N. Tour-based mode choice modelling as the core of an activity-based travel demand modelling framework: a review of state-of-the-art. Transport Rev. 2021, 41, 5-26. DOI: https://doi.org/10.1080/01441647.2020.1780648
  2. Gardner, B.; Lally, P. Modelling Habit Formation and Its Determinants. In: Verplanken, B. (eds) The Psychology of Habit. Springer, Cham, 2018, 207-229. DOI: https://doi.org/10.1007/978-3-319-97529-0_12
  3. Tatsukawa, Y.C.; Arefin, M.R.; Tanaka, M.; Tanimoto, J. Free ticket, discount ticket or intermediate of the best of two worlds – Which subsidy policy is socially optimal to suppress the disease spreading? Theo. Biol. 2021, 520, 110682. DIO: https://doi.org/10.1016/j.jtbi.2021.110682
  4. Guo, Y.T.; Peeta, S.; Agrawal, S.; Benedyk, I. Impacts of Pokémon GO on route and mode choice decisions: exploring the potential for integrating augmented reality, gamifcation, and social components in mobile apps to infuence travel decisions. Transportation 2022, 49, 395-444. DOI: https://doi.org/10.1007/s11116-021-10181-9
  5. Xu, J.X.; Zhang, J.; Guo, J.N. Contribution to the field of traffic assignment: A boundedly rational user equilibrium model with uncertain supply and demand. Socio-Econ. Plan. Sci. 2021, 74, 100949. DOI: https://doi.org/10.1016/j.seps.2020.100949
  6. Wang, X.; Mohcine, C.; Chen, J.; et al. Modeling boundedly rational route choice in crowd evacuation processes. Safety Sci. 2022, 147, 105590. DOI: https://doi.org/10.1016/j.ssci.2021.105590

  1. The authors mentioned in detail in terms of the experiment procedure and participants’ characteristics. However, it is not clear how many participants are there.

Thanks for the reminder, I have checked the whole paper, and make the following modification:

57 participants finished the whole experiment during the two-month experimental window.

  1. What are the limitations of this study?

Thanks for the reminder. In the revision paper, we have added the following contents in conclusions section

It is worth mentioning that due to the difference between experimental design and actual situation, the scope of this study is restricted to an awareness of the conflict between car commuters’ route choice habitual behaviors and traffic information search behaviors. Even so, there are also sone limitations in this study, for example, the measure of traffic information’s compliance ratio is preliminary, and this makes the compliance ratio under the en-route traffic information may be higher than reality; moreover, the compliance ratios under the pre-trip traffic information and en-route traffic information are also higher than they actually is. Thus, in future research, it is necessary to build a reasonable and effective method to calculate drivers’ information compliance ratio.

Reviewer 2 Report

The reviewed article contains interesting considerations. The extensive practical analysis of issues related to the analysis on the conflict between car commuter’s route choice habitual behavior and traffic information search behavior, but the current version need improved, for details se below:

-        article title: according style guide, nouns, pronouns, verbs, adjectives, and adverbs are the words capitalized in titles of articles,

-        page 4-5: add subsection “3.1. ………………..”,

-        line 499, Conclusions: expand your conclusions,

-        the author need to clarify and explain the difference between the current study with the available literature,

-        line 518-519: complete the description “Supplementary Materials”,

-        page 14: add “Institutional Review Board Statement”, “Data Availability Statement”,

-        References: is required Abbreviated Journal Name,

-        References: include the digital object identifier (DOI) for all references where available,

-        References: please format according to the journal guidelines
https://www.mdpi.com/files/word-templates/sensors-template.dot

In future publications, author should devote more time to editing the article according to the requirements of the journal. This will then avoid quite a number of insights into editing.

The reviewed article is a valuable publication. It can serve readers as a set of knowledge that can be used as a basis for further innovative and implementation studies.

Author Response

Thanks for the reminder and suggestion. In the revision paper, I have made the following modification:

  1. The article title: Analysis on the Conflict Between Car Commuter’s Route Choice Habitual Behavior and Traffic Information Search Behavior

  1. Added subsection “1. Experimental design”

  1. Expand our conclusions. In the revision paper, I have made the following modification:

This study focuses on the conflict between car commuters’ route choice habitual behaviors and traffic information search behaviors. An experiment was conducted to obtain participants’ route choice data as well as traffic information search data. Then, different methods had been applied to deal with the experimental data and came to the following conclusions: 1) at the beginning, the traffic information had great influence on car commuters’ route choice habitual behaviors; but as time goes on, the effect of traffic information was on the decline. 2) the different way of traffic information search behaviors (i.e., actively or passively), could have different influence on habit participants’ response times and cause different information compliance ratios. 3) habit will reappear in car commuters’ daily route choices when they get well acquainted with the traffic information. 4) policies for evaluating the impact of traffic information should consider commuters’ habitual behavior, their traffic information search behaviors, and focus on both short-term and long-term.

  1. Clarify and explain the difference between the current study with the available literature. In the revision paper, I have made the following modification:

To answer the above question, this paper focuses on the car commuters’ daily route choices (for most car commuters, the daily commuting happens in a relatively stable situation with fixed origin-destination pair, fixed stare working time, and similar road condition, if the extreme weather, traffic accident, and life events are not considered), and to explore the conflict between car commuters’ daily route choice habitual behaviors and traffic information search behaviors. To our best knowledge, focus on the psychology of the habit (e.g., automatically responding, repeated behavior, and without cognitive elaboration), little researches shed light on the relationship between habit travelers’ route choices and their traffic information search behaviors, and expose the impact of traffic information on habit travelers’ daily route choices in short-term and in long-term, respectively.

  1. Question about “Supplementary Materials”.

Thank you for the reminder! All the materials (Figures and Tables) have been presented in this paper. There is no material to supplement. So, the “Supplementary Materials” has been removed in the revision paper.

  1. In the revision paper, we have added “Institutional Review Board Statement”, “Data Availability Statement”.

  1. Question about “References”. In the revision paper, the journals’ names have been abbreviated.

  1. In the revision paper, we have added the digital object identifier (DOI) for all references.

Reviewer 3 Report

I highly appreciate the reviewed article. 

It presents an analysis of the behaviour of drivers commuting in their own cars, taking into account the use of information on traffic density along the route.

I conclude that:

Firstly, the Author has followed the literature on the issue very carefully. 

Secondly, the research conducted by the Author required a great deal of personal commitment on his part. I know, because I myself took part in similar research in the field of psychology of drivers' behaviour on the road.

Thirdly, the analysis of the research results is carried out correctly with the use of statistical analysis.

Fourthly, the description of the results is very well done. The article reads very well.

Finally, the conclusions are interesting not only from the scientific point of view but also from the practical point of view.

I, therefore, believe that this article should be published without delay.

Author Response

Thank you for your affirmation to my research works. Finally, thanks for your review and comment.

Reviewer 4 Report

The article titled Analysis on the conflict between car commuter’s route choice habitual behavior and traffic information search behavior is a well-structured and clearly written article. The methodology is well described and its limitations are clearly highlighted.

In the continuation of the research, the collection of data in real traffic conditions would give greater reality to the research, but the conducted experiment provides valuable data that will have a significant impact in designing the continuation of the research.

I ask the authors to comment on whether during their research they noticed potential correlations between, for example, the age or level of education of the respondents on the strength of the habit and the influence of traffic information on the choice of route.

It would be interesting to analyze the influential parameters and make a prediction model for real traffic conditions. In doing so, it is important to clarify that on-site information will be processed if the driver does not have a traffic jam or complex traffic situation.

The driver's sense of safety when choosing a route (or changing the route) proved to be an important input parameter (complexity of intersections, traffic regulation, road slopes in winter conditions, etc.), so I would recommend that the authors briefly discuss the impact of transport infrastructure and traffic regulation measures on flexibility traffic users in route selection.

Author Response

Thank you for your review and comments.

  1. In our research, we find that there are no significant age or education difference in participants. Due to the difference between experimental design and actual situation, and limit of the sample participants, we mainly focus on studying the conflict between car commuters’ route choice habitual behaviors and traffic information search behaviors.

  1. Yes, it would be interesting to analyze the influential parameters and make a prediction model for real traffic conditions.

For example:

Assumption 1: With traffic information, the no habit participants are utility maximizers. No habit participants would choose route  in Round  with a probability of , as in Eq.:

Where  is the perceived travel time of route  in Round

As stated by Eq. , when given traffic information, no participants would choose the route with the shortest perceived travel time. It is further assumed that  follows an independent and identical Gumbel distribution. Then, the probability  can be calculated as in Eq.:

where  is the perception parameter. The smaller the value of , the larger the variance in perception error.

Assumption 2: With traffic information, a habit participant sticks to his/her current route with a probability of  (Eq.) if he/she chooses route  in Round :

where  is the inertial threshold of habit participants. Cantillo and Ortúzar (2006) defined inertial threshold as the minimum perceptible change and argued that “attribute changes below the threshold do not cause a reaction in the individual.” In this study,  was calculated as the difference in actual travel times between the 2 alternative routes in Round : , . The captive participant would switch from his/her current route  to the alternative route with a probability of .

Similarly, if  is assumed to be independently and identically Gumbel-distributed, then probability  can be further calculated as the following logit formulation (Cantillo et al., 2007):

where  is the perception parameter. The smaller the value of , the larger the variance in perception error.

not have a traffic jam or complex traffic situation

1)     Participants with no habit choice behavior, but have traffic information search behavior

2)     Participants with no habit choice behavior, and also have no traffic information search behavior

3)     Participants with habit choice behavior, but have no traffic information search behavior

4)     Participants with habit choice behavior, and also have traffic information search behavior

have a traffic jam or complex traffic situation

5)     Participants with no habit choice behavior, but have traffic information search behavior

6)     Participants with no habit choice behavior, and also have no traffic information search behavior

7)     Participants with habit choice behavior, but have no traffic information search behavior

8)     Participants with habit choice behavior, and also have traffic information search behavior

The research scenario is complicated. To do this, we need more participants. Thus, we can get enough data to estimate values of β_1 and β_2

Our experiments do not support such research. However, it is interesting work, can give us a lot of enlightenment.

  1. Thanks for the suggestion. In the revision paper, I have added the following contents in discussion

Round 2

Reviewer 1 Report

The authors have addressed all my concerns.